# Modified World Health Organization (WHO) Tunnel Test for Higher Throughput Evaluation of Insecticide-Treated Nets (ITNs) Considering the Effect of Alternative Hosts, Exposure Time, and Mosquito Density

**DOI:** 10.3390/insects13070562

**Published:** 2022-06-21

**Authors:** Dismas S. Kamande, Olukayode G. Odufuwa, Emmanuel Mbuba, Lorenz Hofer, Sarah J. Moore

**Affiliations:** 1The Nelson Mandela African Institution of Science and Technology (NM-AIST), Arusha P.O. Box 447, Tanzania; smoore@ihi.or.tz; 2Vector Control Product Testing Unit (VCPTU), Ifakara Health Institute, Environmental Health, and Ecological Sciences, Bagamoyo P.O. Box 74, Tanzania; oodufuwa@ihi.or.tz (O.G.O.); embuba@ihi.or.tz (E.M.); 3MRC International Statistics and Epidemiology Group, London School of Hygiene and Tropical Medicine, London WC1E 7HT, UK; 4Vector Biology Unit, Department of Epidemiology and Public Health, Swiss Tropical & Public Health Institute, Kreuzstrasse 2, 4123 Allschwil, Switzerland; lorenz.hofer@swisstph.ch; 5Faculty of Science, University of Basel, Petersplatz 1, 4001 Basel, Switzerland

**Keywords:** WHO tunnel test, insecticide treated nets, ITNs, interceptor, interceptor G2, membrane, human arm, rabbit, bioassay, bio-efficacy, mosquito, *Anopheles*

## Abstract

**Simple Summary:**

Membrane feeding assays have been widely used in malaria transmission research and insectary colony maintenance. Here, we investigate whether a membrane feeder can replace animal baits for evaluating insecticide-treated nets (ITNs) bio-efficacy in the World Health Organization (WHO) tunnel test. The effect of (1) alternative baits, (2) exposure time, and (3) mosquito density on the endpoints of mosquito mortality and feeding inhibition or feeding success was investigated. Our results show that similar mortality at 24-h (M24) or 72-h (M72) is estimated using either a membrane feeder or a rabbit bait with an overnight (12 h) exposure. However, the membrane measured higher blood feeding inhibition than the rabbit, likely due to the absence of host cues, notably carbon dioxide. Therefore, the membrane feeder may be used instead of an animal bait to test mortality endpoints in WHO tunnel tests and blood feeding rates need to be improved. Experimental results demonstrated that using 50 or 100 mosquitoes per replicate measure the same for mortality and feeding inhibition endpoints with an animal bait. Therefore, WHO tunnel tests may be run with lower mosquito densities. This will reduce strain on insectaries to produce sufficient mosquitoes to meet the large sample sizes needed for bio-efficacy durability monitoring of chlorfenapyr ITNs that must be evaluated in “free-flying” bioassays.

**Abstract:**

The standard World Health Organization (WHO) tunnel test is a reliable laboratory bioassay used for “free-flying” testing of insecticide-treated nets (ITNs) bio-efficacy where mosquitoes pass through a ITN sample to reach a live animal bait. Multiple parameters (i.e., bait, exposure time, and mosquito density) may affect the outcomes measured in tunnel tests. Therefore, a comparison was conducted of alternative hosts, exposure time, and lower mosquito density against the current gold standard test (100 mosquitoes, animal bait, and 12-h exposure) as outlined in the WHO ITN evaluation guideline. This was done with the aim to make the tunnel test cheaper and with higher throughput to meet the large sample sizes needed for bio-efficacy durability monitoring of chlorfenapyr ITNs that must be evaluated in “free-flying” bioassays. **Methods:** A series of experiments were conducted in the WHO tunnel test to evaluate the impact of the following factors on bio-efficacy endpoints of mosquito mortality at 24-h (M24) and 72-h (M72) and blood-feeding success (BFS): (1) baits (rabbit, membrane, human arm); (2) exposure time in the tunnel (1 h vs. 12 h); and (3) mosquito density (50 vs. 100). Finally, an alternative bioassay using a membrane with 50 mosquitoes (membrane-50) was compared to the gold standard bioassay (rabbit with 100 mosquitoes, rabbit-100). Pyrethroid-resistant *Anopheles arabiensis* and pyrethroid susceptible *Anopheles gambiae* were used to evaluate Interceptor^®^ and Interceptor^®^ G2 ITNs. **Results:** Using a human arm as bait gave a very different BFS, which impacted measurements of M24 and M72. The same trends in M24, M72 and BFS were observed for both Interceptor^®^ ITN and Interceptor^®^ G2 unwashed and washed 20 times measured using the gold standard WHO tunnel test (rabbit-100) or rabbit with 50 mosquitoes (rabbit-50). M24, M72 and BFS were not statistically different when either 50 or 100 mosquitoes were used with rabbit bait in the tunnel bioassay for either the susceptible or resistant strains. No systematic difference was observed between rabbit-50 and rabbit-100 in the agreement by the Bland and Altman method (B&A). The mean difference was 4.54% (−22.54–31.62) in BFS and 1.71% (−28.71–32.12) in M72 for rabbit-50 versus rabbit-100. Similar M24, M72 and lower BFS was measured by membrane-50 compared to rabbit-100. No systematic difference was observed in the agreement between membrane-50 and rabbit-100, by B&A. The mean difference was 9.06% (−11.42–29.64) for BSF and −5.44% (−50.3–39.45) for M72. Both membrane-50, rabbit-50 and rabbit-100 predicted the superiority of Interceptor^®^ G2 over Interceptor^®^ ITN for the resistant strain on M72. **Conclusion:** These results demonstrate that WHO tunnel tests using rabbit bait may be run with 50 mosquitoes to increase sample sizes needed for bio-efficacy durability monitoring of ITNs in “free-flying” bioassays. Using a membrane feeder with 50 mosquitoes is a potential replacement for the WHO tunnel bioassay with animal bait if control blood feeding rates can be improved to 50% because blood feeding impacts mosquito survival after exposure to insecticides.

## 1. Introduction

Vector control continues to offer effective prevention of mosquito-borne disease globally [1]. Insecticide-treated nets (ITNs) have been an extremely effective control measure [2] because they interrupt malaria transmission in two ways, by reducing mosquito blood-feeding and by killing a proportion of mosquitoes that contact the nets [3,4]. Since 2015, however, malaria control progress has stalled, with the COVID-19 pandemic in 2020 placing additional constraints on malaria control efforts. Despite this, ITNs remain the current cornerstone of global malaria control [1].

To date, all ITNs contain pyrethroid insecticides, which reduce the number of bites that individuals sleeping under them receive even if the nets become old and torn, because pyrethroids inhibit mosquito flight and feeding responses [5,6]. However, mass deployment of pyrethroid ITNs globally has led to widespread pyrethroid resistance with varying mechanisms [7] observed in eighty two countries. To sustain the malaria control gains attributed to ITNs and to assist in reducing malaria by at least 90% by 2030 [8], several ITNs with different insecticide classes in combination with pyrethroids have been developed. These so-called “dual-insecticide ITNs’’ afford non-neurotoxic modes of action with no cross-resistance (chlorfenapyr), reduced fecundity, and fertility (pyriproxyfen, PPF), or increased susceptibilities to pyrethroids (piperonyl butoxide, PBO) [9,10,11,12,13,14]. Randomized control trials have demonstrated greater malaria control using dual-active ITNs compared to pyrethroid nets in areas of high pyrethroid resistance, with pyrethroid combined with PBO [15,16], or chlorfenapyr [17]. Operational research has indicated an additional public health benefit of chlorfenapyr [18] and pyriproxyfen [18] in combination with pyrethroid compared to pyrethroid-only ITNs.

New ITN products must demonstrate their continued effectiveness for malaria control up to three years after deployment through biological efficacy testing against mosquito vectors [19,20]. The current World Health Organization (WHO) guidelines for ITN testing outline bioassays that were designed to evaluate pyrethroids with rapid neurotoxic action against exposed mosquitoes, i.e., rapid incapacitation (knockdown), reduction in blood-feeding, and killing within 24 h post-exposure. Chlorfenapyr requires the mosquito to be metabolically or physiologically active (as it is when encountering ITNs during host-seeking) to metabolize the parent molecule into the potent n-dealkylated form that elicits mosquito mortality [21]. Mosquitoes are more metabolically active at night, when flying, host-seeking or active during their typical circadian rhythms, for which the “free-flying” WHO tunnel test is a more appropriate bioassay [22,23].

The WHO tunnel test is widely used to assess the bio-efficacy of ITNs under laboratory conditions. Despite predicting a similar bio-efficacy of pyrethroid [5] and chlorfenapyr ITNs [22,23] to those measured in gold-standard experimental hut trials, the tunnel test has several limitations. Firstly, the animal baits (rabbit or guinea pig) used are non-preferred hosts for malaria mosquitoes, especially the highly anthropophilic Afrotropical vectors *Anopheles gambiae*, *Anopheles funestus*, and *Anopheles arabiensis* [24,25]. Moreover, the use of animals includes welfare concerns, and it is costly to ensure that animals are well maintained under veterinary supervision. Secondly, the bioassay is conducted overnight for 12–15 h. There is evidence that mosquitoes interact with treated netting within the first 30 min of release [26], thus prolonging exposure time could overestimate outcomes. Thirdly, the current tunnel test uses one hundred mosquitoes per replicate, which is expensive for insectaries to produce. Owing to the significance of blood-feeding in the life cycle of the malaria mosquito, as well as its importance for malaria transmission between human and mosquito hosts, it is an important component of vector control product testing. Different baits could be used to minimize the limitations of using humans or live animal hosts. Membrane feeders have been widely deployed for evaluating topical mosquito repellents [27], transmission-blocking drugs and vaccines [28], and endectocides [29], as well as for mosquito rearing [30,31,32,33,34,35,36]. Moreover, the use of an artificial membrane has several advantages, including no animal welfare or ethical concerns, reduced chance of accidental disease transmission, simple logistics, and reproducibility [33,35,36,37]. Given the significance of host kairomones in encouraging mosquito feeding, worn socks may be added to augment the attractiveness of the membrane to mosquitoes [38].

Multiple parameters including bait [25], mosquito density [39], and duration of exposure to ITNs [40] may affect the outcomes measured in tunnel tests. Therefore, the current paper compared alternative baits, exposure times, and lower mosquito densities against the current gold standard test (100 mosquitoes, animal bait, and 12 h exposure) as outlined in the WHO ITN evaluation guideline [20] in an attempt to simplify the tunnel test to make it cheaper with a higher throughput for the evaluation of large numbers of ITNs as needed for the bio-efficacy durability monitoring of chlorfenapyr ITNs that must be evaluated in “free-flying” bioassays [41].

## 2. Materials and Methods

### 2.1. Study Area

Bioassays were performed at the Vector Control Product Testing Unit (VCPTU) facility located at the Bagamoyo branch of Ifakara Health Institute (IHI), Tanzania (6.446° S and 38.901° E).

### 2.2. Description of Investigational ITNs

Interceptor^®^ is made from 100-denier polyester coated with 200 mg/m^2^ alpha-cypermethrin and Interceptor^®^ G2 is made of 100-denier polyester coated with a mixture of 200 mg/m^2^ chlorfenapyr and 100 mg/m^2^ alpha-cypermethrin. Both net brands are manufactured by BASF, Germany. Safi Net, made of polyester manufactured by A to Z Textile Mills, Tanzania, was used as a negative control to monitor the quality of the bioassay. The study included the following arms: (1) unwashed Interceptor^®^; (2) Interceptor^®^ washed 20 times; (3) unwashed Interceptor^®^ G2; (4) Interceptor^®^ G2 washed 20 times; (5) negative control–Safi Net. Five samples per net were cut and samples were washed twenty times according to a protocol adapted from the standard WHO washing procedure [20] using 20 g/L palm soap (Jamaa brand). The interval of time used between two washes (i.e., regeneration time) was 1 day for both Interceptor^®^ G2 and Interceptor^®^ ITNs (Table 1).

### 2.3. Mosquitoes

Pyrethroid-resistant *Anopheles arabiensis* (Kingani strain, established 2017) and pyrethroid susceptible *Anopheles gambiae* (Ifakara strain, established 1996) were used in this study. *An. arabiensis* (Kingani) is metabolic-resistant and expresses the upregulation of cytochrome p450s, with 14% mortality upon exposure to WHO 1x discriminating dose of alpha-cypermethrin that was reversed by piperonyl butoxide (PBO) pre-exposure, reconfirmed before this study was initiated. *An. gambiae* s.s. (Ifakara) is fully susceptible to all insecticide classes at 1x WHO discriminating doses, reconfirmed before this study was initiated. The mosquito colony was maintained according to MR4 Guidelines [36] at 27 ± 2 °C and 40%–100% relative humidity, with an ambient (approximately 12:12) light–dark cycle. The colony was maintained on a Tetramin fish food for larvae, 10% glucose for adults. Females were offered cattle blood in a membrane feeder or were offered a human arm as a blood source. Mosquitoes were 5–8 days old, nulliparous, sugar starved for eight hours, and acclimatized to the test room for an hour before the experiment (Table 1). As VCPTU do not have resistant *An. gambiae* in the colony, we used metabolic resistant *An. arabiensis* instead. Since the bioassay measured contact toxicity, it was deemed that the mechanism for resistance was more critical than the species used for the evaluation. 

### 2.4. The Standard WHO Tunnel Test Procedure 

WHO tunnel tests were conducted following WHO guidelines [20] (Figure 1A). The tunnel was divided into two chambers separated by a netting sample that were deliberately holed with 9 small (1 cm) holes through which the mosquitoes had to pass to reach the bait. The bait was placed in the short chamber. In the long section, 100 unfed female mosquitoes aged 5–8 days were released at 19:00 h. The tunnel was covered with a black cloth and left overnight. The following morning, between 07:00 and 09:00 h, mosquitoes were removed from the tunnel using an aspirator. Mosquitoes were scored as alive fed, alive unfed, dead fed, or dead unfed in each chamber and put into a separate paper cup for post exposure mortality monitoring. Mosquitoes were supplied with access to 10% sugar solution *ad libitum* and were then scored for post-exposure delayed mortality at 24-h and 72-h. The experiment and post exposure holding was conducted at a temperature of 27 ± 2 °C and a relative humidity of 80% ± 10. For the experiment to be considered valid, the following thresholds were used: control 24-h mortality ≤10% in all experiments and blood-feeding success ≥50% with experiments using the rabbit bait.

### 2.5. Bait Used and Preparation

**Rabbit:** three groups of five healthy rabbits were used. Rabbits were maintained under veterinary supervision. The rabbit was shaved on its back to allow the mosquitoes to feed. The rabbit was gently restrained in a mesh tube that was suspended in the short section of the WHO tunnel throughout the 12-h experiment (Figure 1B). **Membrane feeding:** A Hemotek^®^ membrane feeder (SP-6 System, Hemotek Ltd., Blackburn BB6 7FD, UK) was used. Two membrane feeders were placed on top of the “bait chamber” of each tunnel (Figure 1C). Each feeder reservoir contained 3 mL of cow blood covered by a stretched parafilm membrane and tightened with an o-ring to prevent any leakage. Cow blood was obtained from cattle maintained under veterinary supervision at VCPTU and was stored for up to two weeks at 4–8 °C in heparinized tubes. Socks worn by the investigator (DK) for 8 h on the day of testing were stretched across the surface of the membrane feeder reservoir to provide host kairomones and increase mosquito attraction to the feeder. The Hemotek^®^ was switched on 10 min before the experiment. The temperature of the feeder was set at 37–39 °C throughout the 12-h experiment. **Human arm:** Five healthy male volunteers conducted arm feeding by inserting their arms into the bait short section of the tunnel (Figure 1D). Before testing, their arms were washed with water. The volunteers were non-smokers and did not drink alcohol or use perfumed lotions during the experimental period. The experimental time for arm feeding was 1 h to allow for standardized evaluation and to minimize volunteer discomfort. Previous work has shown that 30 min of exposure resulted in high blood feeding [35]. To protect human participants, several procedures are routinely undertaken in the laboratory. Anybody who works in the insectary and blood-feeds mosquitoes (including the participants) are screened weekly for malaria parasites using malaria rapid tests (SD bioline). Colony mosquitoes are not kept beyond 10 days, as it takes 12–14 days for mosquitoes to develop sporozoites. Mosquitoes used in the experiments were nulliparous. Therefore, participants were not at risk of malaria infection as a result of the experiments. 

### 2.6. Study Design

Experiments were comparative bioassays with a minimum of 5 replicates per net type, per permutation (Table 1). A total of sixty one experimental nights were run between March 2021 and February 2022. All procedures for preparation, release, collection, and mosquito scoring were performed as per the standard WHO tunnel test procedure [20] (Figure 1A) outlined above with the factors of interest (bait, exposure time, and density) varied (Table 1). The endpoints measured were blood feeding success (BFS) or blood feeding inhibition (BFI), mortality at 24-h (M24), and mortality at 72-h (M72).

#### 2.6.1. Experiment 1: The Impact of Bait/Host

The bio-efficacy of unwashed and 20 times washed Interceptor^®^ G2 and Interceptor^®^ ITNs was tested using 100 pyrethroid-resistant *An. arabiensis* per replicate with membrane, human arm, and rabbit bait (Figure 2A). Mosquitoes were left in the tunnel for 12 h overnight and BFS, M24, and M72 endpoints were evaluated. Five samples for each ITN type and condition (Interceptor^®^ G2 unwashed and 20× washed and Interceptor^®^ unwashed and 20× washed) for each host type were evaluated using five tunnels. One control and four treatments—i.e., one per net type and condition—were conducted each night for 15 nights with each bait (membrane, human, and rabbit) and were evaluated for five nights each. Each bait type was tested on different nights to allow for an independent comparison of each bait in the absence of competing host kairomones. 

#### 2.6.2. Experiment 2: The Impact of Exposure Time

The bio-efficacy of unwashed and 20× washed Interceptor^®^ G2 and Interceptor^®^ was tested using 100 pyrethroid-resistant *An. arabiensis* per replicate with either a human arm or membrane bait (Figure 2B). When investigating 1 h exposure, mosquitoes were exposed to ITNs for only 1 h with a human arm or membrane and were then removed from the tunnel and placed in holding cups with access to sugar for 11 h overnight. For the 12-h exposure, the human arm was only available for 1 h, but the mosquitoes were left in the tunnel for the remaining 11 h of the experiment. In the membrane assay, mosquitoes interacted with membrane in the tunnel throughout the 12 h of exposure. In both tests, the BFS, M24, and M72 endpoints were evaluated. Five samples for each ITN type (Interceptor^®^ G2 unwashed and 20× washed and Interceptor^®^ unwashed and 20× washed) plus a negative control were tested using five tunnels. Five replicates per treatment arm for each bait and exposure time were conducted over 10 nights. The 1 h and 12 h of exposure were conducted on the same night for either the membrane or the human arm. The 1 h exposure was performed and then a 12-h exposure was conducted on the same net using a fresh batch of mosquitoes. Each bait type was tested on different nights to allow for an independent comparison of each bait in the absence of competing host kairomones.

#### 2.6.3. Experiment 3: Effects of Mosquito Density on the Bio-Efficacy Measurement of Blood-Feeding Inhibition and Mortality at 24-h or 72-h

The effect of mosquito density on bio-efficacy measurements of BFS, M24, and M72 endpoints was evaluated in the WHO tunnel using 50-mosquitoes compared to the standard 100-mosquitoes (Figure 3A). Experiments were conducted following standard procedures with 12 h of exposure and continuous access to a restrained rabbit. For this, two species were used: pyrethroid-resistant *An. arabiensis* tested for the pyrethroid and chlorfenapyr Interceptor^®^ G2 (unwashed or 20× washed) and pyrethroid-susceptible *An. gambiae* for the pyrethroid only Interceptor^®^ ITN (unwashed or 20× washed). A total of seven tunnels (one control, 3 with unwashed, and 3 with washed ITNs) per night were run with 15 replicates conducted per net condition for each mosquito density. Each strain and density (Table 1) were evaluated in a separate 5-night block. This was done to ensure the fitness of mosquitoes used, as the experiments were conducted at a time when the mosquito colony was under pressure from multiple evaluations.

#### 2.6.4. Experiment 4: Possibility to Replace Standard Bait (Rabbit) with the Membrane Assay

To determine whether the rabbit can be replaced with the membrane assay as the bait, the bio-efficacy measurements of BFS, M24, and M72 endpoints were evaluated in the WHO tunnel with 12 h of exposure using 50-membrane and 100-rabbit (gold standard) with resistant *An. arabiensis* mosquitoes (Figure 3B). The same procedure was used for all five arms: a negative control and four treatment arms of Interceptor^®^ unwashed or 20× washed and Interceptor^®^ G2 unwashed or 20× washed (Table 1). For the membrane, a total of 5 tunnels (1 per arm) were run per night, and for the rabbit, 9 tunnels (1 control and 2 replicates per treatment arm) were run per night, with a total 15 replicates per arm for each assay. Different baits were run on separate nights to allow for an independent comparison of each bait in the absence of competing host kairomones.

### 2.7. Data Analysis

#### 2.7.1. Sample Size and Power

A sample size calculation for generalized linear mixed effects models (GLMMs) through simulation [42] in R statistical software 3.02 https://www.r-project.org/ (accessed on 23 April 2022) was performed to detect a 10% effect difference between the nets, simulations were performed using an estimated mosquito mortality of 80% for unwashed Interceptor^®^ G2 and 70% for unwashed Interceptor^®^, and 10% for SafiNet^®^ (deliberately holed). The power estimated was more than 90% based on estimates from previous studies conducted in the same setting: mean mortality of 81.5% for the WHO tunnel test with an assumed daily variation of 0.5 and 15 replicates per arm [23].

#### 2.7.2. Statistical Analysis

Data were collected using standard paper forms and double entered into an Excel spreadsheet, cleaned, and imported into STATA 16.1 (Stata Statistical Software: Release 16. College Station, TX, USA: StataCorp LLC.) for analysis. Descriptive statistics were used for data summarization, whereby mean percentage mortality at 24-h (M24) or 72-h (M72) or blood feeding success (BFS) or blood-feeding inhibition (BFI) with their 95% Confidence Intervals (CI) were calculated. Multivariable mixed logistic regression with a binomial link was conducted with fixed effects for the exposure of interest, adjusting for ITN condition and mosquito species, with day as a random effect to account for daily variability in environmental conditions and mosquito batch variability. Model fit was checked by the testing of model residuals. To estimate the superiority of Interceptor^®^ G2 over Interceptor^®^ with resistant mosquitoes, the same regression was used for comparing superiority measured using the gold standard 100-rabbit to 50-membrane on M72 and the BFS endpoint. In addition, Bland and Altman [43] methods were used to estimate the agreement in outcomes M24, M72 and BFS measured by assays: (1) membrane vs. rabbit; (2) 100 vs. 50 mosquitoes; and (3) 100-rabbit vs. 50-membrane.

## 3. Results

### 3.1. Experimental Validity 

In all the bioassays conducted, control M24 was <10% and at M72 was <13%. BFS was ≥50% in both the human arm and the rabbit controls and was <23% in the membrane control (Table 2).

### 3.2. Experiment 1: The Impact of Baits

The bait used affected both the feeding and mortality endpoints measured. The membrane measured a similar mortality and a lower blood feeding success than the rabbit. The human arm measured a lower mortality and higher blood feeding success than the rabbit.

M24 in the intervention arms was not significantly different between the rabbit and membrane (OR: 0.90, 95% CI: 0.79–1.02, *p* = 0.086) and was significantly lower using the human arm (OR: 0.42, 95% CI: 0.37–0.48, *p* < 0.001) compared to the rabbit (Table 2). M72 in the intervention arms was not significantly different between the rabbit and membrane (OR: 1.07, 95% CI: 0.93–1.22, *p* = 0.352) and was significantly lower using the human arm (OR: 0.31, 95% CI: 0.27–0.35, *p* < 0.001) compared to the rabbit (Table 2). Control M24 was higher in the membrane and human arms; but control M72 was higher in the human arm (OR: 1.83, 95% CI: 1.22–2.75, *p* = 0.004) and was not different between rabbit and membrane (OR: 1.16, 95% CI: 0.84–1.59, *p* = 0.366). In the treatment arm, BFS was significantly lower using a membrane (OR: 0.34, 95% CI: 0.28–0.48, *p* < 0.001) and was significantly higher using a human arm (OR: 9.81, 95% CI: 8.25–11.67, *p* < 0.001) compared to the rabbit (Table 2). The same trend was observed in the control arm. 

The same trends in mortality and blood feeding inhibition (BFI) were observed for both Interceptor^®^ ITN and Interceptor^®^ G2 (Figure 4). Higher blood feeding resulted in lower mortality (Appendix A), which will explain the lower mortality measured with the human arm, which also had substantially higher BFS. Therefore, the human arm could not be considered for further evaluation. Between the membrane and the rabbit with 100 mosquitoes per replicate, no systematic difference was observed for agreement by Bland and Altman methods (Appendix A). The mean difference was 6% (−10.81–23.01) for BFS and −1.09% (−72.91–70.73) for M72.

### 3.3. Experiment 2: Impact of Exposure Time on Mortality and Blood Feeding 

Increasing the time that mosquitoes are left in the tunnel from 1-h to 12-h increased mortality with either the human arm or the membrane (Table 3). With the membrane bait, longer exposure significantly increased both the odds of M72 (OR: 2.30, 95% CI: 2.02–2.62, *p* = 0.001) and the odds of BFS (OR: 1.55, 95% CI: 1.08–2.22, *p* = 0.017). Similarly, in the human arm, the longer exposure significantly increased the odds of M72 (OR: 1.66, 95% CI: 1.45–1.90, *p* = 0.001), while the effect of exposure time on BFS could not be measured since the human arm was only available for one hour (Figure 5). The time that mosquitoes are left in the tunnel overnight is a significant factor in mosquito mortality and should always be recorded and reported.

### 3.4. Experiment 3: Effects of Mosquito Density on Tunnel Test Endpoints

M24, M72, and BFS were very similar and were not statistically different when either 50 or 100 mosquitoes were used in the tunnel bioassay with rabbit bait for either the susceptible or resistant strains (Table 4). This was consistent for both Interceptor^®^ and Interceptor^®^ G2, unwashed and washed 20 times (Figure 6). No systematic difference in agreement between methods was observed by Bland and Altman methods (Appendix A). The mean difference was −4.54% (−31.62–22.54) in BFS and 1.71% (−28.71–32.12) in M72. Furthermore, when tested using the pyrethroid-resistant strain, the 50-rabbit bioassay predicted the superiority of Interceptor^®^ G2 to Interceptor^®^, as did the 100-rabbit (Table 5). 

However, when considering the superiority of Interceptor^®^ and Interceptor^®^ G2, the lower mosquito density (50) resulted in a higher BFS in the Interceptor^®^ G2 arm (Table 5). This indicates that mosquitoes at a high density are either interacting with each other to disturb each other from feeding, or discomfort from high biting rates is making the host more defensive. This increased blood feeding success is likely translating into the lower odds of mortality observed for washed Interceptor^®^ G2 relative to Interceptor^®^ using 50 mosquitoes (OR: 1.07, 95% CI: 0.85–1.34, *p* = 0.579) compared to 100 mosquitoes (OR: 1.31, 95% CI: 1.12–1.54, *p* = 0.001) (Table 5). This observation underlines the importance of consistent control blood feeding success on mortality estimates from the WHO tunnel test and this should always be recorded and reported. 

### 3.5. Experiment 4: Possibility to Replace Standard Bait with the Membrane Feeding

The membrane assay with 50 mosquitoes (membrane-50) did not measure statistically different M24 or M72 compared to the rabbit with 100 mosquitoes (rabbit-100) (Table 6) when testing pyrethroid only Interceptor^®^ or Interceptor^®^ G2 against pyrethroid-resistant *An. arabiensis.* Again, BFS was different, with a far higher BFS in the rabbit-100 assay than in the membrane 50-assay. 

However, when used for predicting the difference in bio-efficacy between Interceptor^®^ and Interceptor^®^ G2, both assays were measured in the same way (Figure 7) and both predicted superior odds of M72 for Interceptor^®^ G2 (100-rabbit OR: 1.23 (95% CI: 1.10–1.38), *p* < 0.0001; 50-membrane 1.79 (95% CI: 1.50–2.14) *p* < 0.0001) and inferior reduction in blood feeding (100-rabbit OR: 1.76 (95% CI: 1.47–2.10), *p* < 0.0001; 50-membrane 1.87 (95% CI: 1.05–3.33) *p* = 0.033) with Interceptor^®^ G2 relative to Interceptor^®^ (Table 7). No systematic difference was observed in agreement for membrane-50 and rabbit-100 by Bland and Altman methods, with a mean difference (and limits of agreement) of 9.06 % (−11.42–29.54) on BFS and −5.43 % (−50.3–39.45) on M72 (Appendix A).

## 4. Discussion

The current tunnel test uses one sample of ITN with 100 mosquitoes as the unit of replication and based on the current work, it is proposed that a larger number of nets or two samples per ITN can be tested using 50 mosquitoes per replicate to improve laboratory throughput. Biological durability monitoring requires large sample sizes, as nets are exposed to highly variable use patterns [44,45,46,47,48] and environmental conditions [49,50] that result in a high degree of heterogeneity between individual nets. The goal of biological durability monitoring is the precise estimation of the biological efficacy of a population of ITNs. As the ITN is the unit of replication, greater precision is obtained by evaluating larger numbers of ITNs. 

The current experiment confirmed that using the Hemotek^®^ membrane feeding system as a blood source, together with a worn sock emitting human odor with a replicate size of 50 mosquitoes, a similar mortality and feeding inhibition as the standard WHO tunnel bioassay with rabbit and a replicate size of 100 mosquitoes for both pyrethroid and mixture pyrethroid and chlorfenapyr ITNs is estimated. Our results suggest that a membrane bioassay can evaluate the difference between ITNs because the membrane assay estimates the superiority of Interceptor^®^ G2 over Interceptor^®^ on the M72 outcome using metabolic-resistant mosquitoes, which was also measured by the gold standard rabbit-100 assay and has been consistently seen in other studies in the WHO tunnel, I-ACT, and experimental hut [23]. It was also able to predict the superior blood feeding inhibition of Interceptor^®^, which has a higher concentration of the pyrethroid alpha-cypermethrin (200 mg/m^2^ alpha-cypermethrin in Interceptor^®^ and 100 mg/m^2^ alpha-cypermethrin in Interceptor^®^ G2). Being able to test differences between products is the goal of durability monitoring bioassays that track the bio-efficacy of ITNs over time (biological durability) and compare them to unwashed positive controls [41].

Having a reliable bioassay that can be conducted routinely without animal welfare concerns will be extremely useful. The data generated by the current work are promising and further work is planned to improve mosquito feeding success on the membrane as it was seen that differences in blood feeding success do impact on the mortality estimates. While this did not impact on the predictions of superiority, and therefore mortality can still be compared to an unwashed positive control net, if thresholds are used, i.e., the proportion of nets that meet WHO bio-efficacy criteria, then this might affect the interpretation of the bioassay results. It is recommended that the results are replicated in additional laboratories, since having an assay that can accurately predict the differences between net samples in multiple laboratories with several pyrethroid-resistant mosquito strains and that can predict the results of experimental hut studies is optimal. Data from the experiments demonstrated that several factors influenced the mean mortality and feeding inhibition estimated in WHO tunnel tests [22,51,52].

### 4.1. Impact of the Bait

The use of different baits had an enormous influence on the bioassays. By using a human arm as bait, feeding inhibition was substantially lower compared to membrane or rabbit baits [53]. This has also been seen in early versions of the tunnel test [51]. This preference for the human arm is unsurprising, since the colony used in the experiments is anthropophilic. Therefore, although it is more representative of end-user conditions, the use of a human is not recommended for ITN evaluation, because the results were not comparable to those of the rabbit bioassay that was shown to predict the results of experimental hut trials in this setting [23] and elsewhere [9,22,52]. Study findings using *An. arabiensis* mosquitoes were consistent with the existing literature on vector host preference [25,54], confirming that mosquitoes are most attracted to humans as bait, followed by rabbits, and were least attracted to the membrane. Lower attraction in assays using the Hemotek^®^ membrane system and rabbits reduces the number of mosquitoes passing the ITN tested, resulting in higher feeding inhibition compared to when the human arm was used as bait. Several other studies have shown that host-seeking *An. arabiensis* are more attracted to humans than to live animals [24,25]. The lower attraction and consequent higher feeding inhibition when using a membrane is likely due to the absence of carbon dioxide (CO_2_) that increases mosquito responses to kairomones [55] and the small size of the membrane feeder’s surface, which reduces the amount of heat and moisture available, which are both important short-range attractants to mosquitoes [56,57,58]. The validity of the experiment relies on the negative control feeding success of (>50%) for rabbits. In this assay, with the membrane, augmentation with socks that contained human kairomones improved the attraction of the membrane to mosquitoes [59]. However, it was not possible to use the same threshold value for feeding success with the less attractive membrane. For this reason, further work is needed to optimize the attraction of the membranes for use in the WHO tunnel test. Further improvements to the attractiveness of the membrane could be achieved by making a larger surface area available [60,61] and the addition of 2-butanone [62] or CO_2_ [63] to augment mosquito response to kairomones until 50% feeding success in the negative control is consistently achieved.

### 4.2. Impact of Exposure Time

Exposure time was important with 12 h exposure, increasing both mortality and feeding success, indicating that the mosquitoes make repeated contact with the ITN sample overnight. Consistently, prolonged exposure (12 h) increased the efficacy of insecticide and host-seeking activities compared to 1 h exposure, resulting in increased mortality because of a higher dose of insecticide picked up by the mosquitoes when resting, bouncing, and passing the ITNs repeatedly. This is also likely in experimental huts and in the community where ITNs are in use. Therefore, the use of a 12 h overnight exposure is recommended. For insecticides that require the mosquitoes to be metabolically active, such as chlorfenapyr, prolonging exposure to 12 h allows the conversion of parent molecules into active forms, because of mosquitoes’ metabolic activity when flying in the tunnel. Interestingly, results show that with either the pyrethroid only Interceptor^®^ or the pyrethroid-chlorfenapyr Interceptor^®^ G2 ITNs higher mortality was observed among unfed mosquitoes. Therefore, the results of this study underline the WHO recommendation that feeding success should always be reported when conducting WHO tunnel tests, as low feeding rates will affect the interpretation of results. 

### 4.3. Effects of Mosquito Density

It was observed that the use of 50 or 100 mosquitoes per testing sample with the rabbit bait did not significantly alter the mortality and blood feeding success measured with either resistant *An. arabiensis* or susceptible *An. gambiae* for the pyrethroid only net or the mixture ITNs. These results suggest that fewer mosquitoes can be used in WHO tunnel bioassays and still correctly measure the efficacy of ITNs. As would be expected, with 50 mosquitoes there is a slight increase in blood feeding success and a consequent slight decrease in mortality compared to assays using 100 mosquitoes. Higher feeding success at a lower density is likely due to less competition between mosquitoes on the membrane during host-seeking [35], which may also reduce the host defensiveness of the rabbit [64,65]. Increasing the number of mosquitoes in the tunnel may lead to density-dependent mortality effects of crowding as mosquitoes can disturb each other when at a high density [66]. Our results suggest that regardless of the insecticides on the ITNs tested, mortality was higher among unfed mosquitoes, revealing an impact of blood feeding on increased mosquito resilience to insecticides after a blood meal. A similar study on the effects of bites through permethrin nets shows that successfully fed mosquitoes survive longer than unfed ones [67]. This has been reported for chlorfenapyr, where observed mortality was lower among blood-fed mosquitoes compared to those who were unfed [10]. Blood feeding elevates detoxifying enzymes (glutathione, monooxygenase), which then assist in the detoxification of insecticides [68], although this did not translate into substantially lower bio-efficacy with Interceptor^®^ G2 as upregulation of metabolization converts the parent molecule into the potent n-dealkylated form that elicits increased mosquito mortality [21]. It is also important to report control blood feeding success because unfit colony mosquitoes are less likely to fly and feed, which reduces the likelihood that the mosquitoes contact treated nets [67], nullifying the bioassay.

### 4.4. Study Limitations

The study has several limitations which should be addressed in subsequent work. Firstly, experiments were conducted in a single testing facility. A comparison of the alternative method in multiple laboratories is desirable to ensure the reproducibility of the methods with other mosquito strains. The low feeding success with the membrane technique needs to be overcome, as clearly feeding success impacts mosquito mortality. Ideally, the membrane bioassay will be improved to consistently measure 50% mosquito feeding success at multiple testing facilities. Additionally, two different ITN products from the same manufacturer were used. It could be argued that the evaluation of dual AI nets of pyrethroids with PBO and PPF would also be as relevant, although these are best measured using WHO cone tests as they do not require “free-flying” bioassays for evaluation. Similar experiments conducted by other facilities are recommended to generate further evidence of the range of values and precision of the estimates of mortality and blood feeding inhibition using the 50-rabbit and 50-membrane technique.

## 5. Conclusions

Here, it was demonstrated that using 50 or 100 mosquitoes with the rabbit gives similar results with no systematic bias for both pyrethroid and pyrethroid-chlorfenapyr ITNs. The lower density can be used for the WHO tunnel test when testing pyrethroid Interceptor^®^ and pyrethroid-chlorfenapyr Interceptor^®^ G2. Reducing the number of mosquitoes per test decreases its cost and allows a larger number of net samples to be tested at a time. Larger sample sizes will give greater precision when estimating ITN efficacy since the unit of replication in ITNs testing is the bioassay (cone, tunnel, I-ACT, experimental hut) and not the mosquito within that assay. Furthermore, we provide the first evidence that membrane feeding systems can be used as an alternative to rabbit bait in WHO tunnel assays. Membrane assay shows an excellent comparison to the gold-standard WHO tunnel test on both the mortality and feeding success endpoint for the ITNs tested, although control feeding success is lower due to the lower attraction of the membrane to host-seeking mosquitoes. Using membrane feeding systems instead of rabbits or other animals in WHO tunnel assays resolves the ethical issues concerning animal welfare and makes the tests simpler to perform. Further work to improve the feeding success of the membrane feeding system as a replacement for rabbits in the WHO tunnel test is needed, as mosquito feeding success impacts insecticide induced mortality. 

## Figures and Tables

**Figure 1 insects-13-00562-f001:**
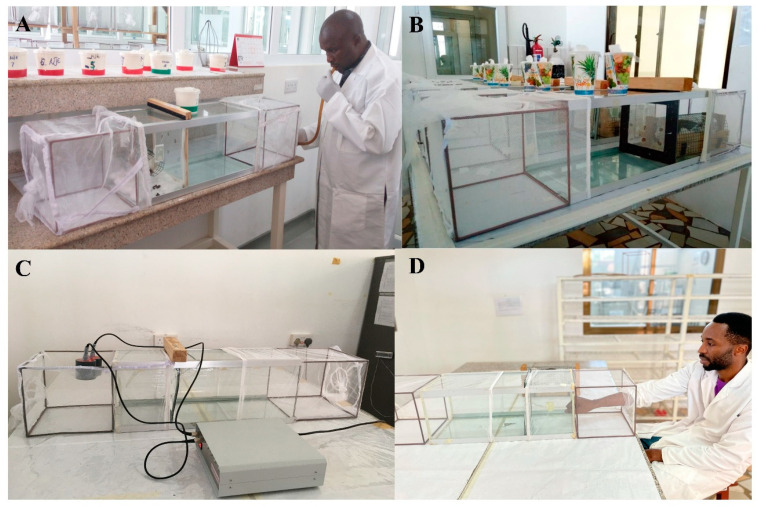
WHO tunnels for comparison of baits: (**A**) Conduct of standard WHO Tunnel with the bait chamber to the left of the picture and mosquitoes being placed into the longer end of the chamber; (**B**) Rabbit—in Experiments 1–4; (**C**) Hemotek^®^ membrane—in Experiment 1 and 4; and (**D**) Human arm—in Experiment 1.

**Figure 2 insects-13-00562-f002:**
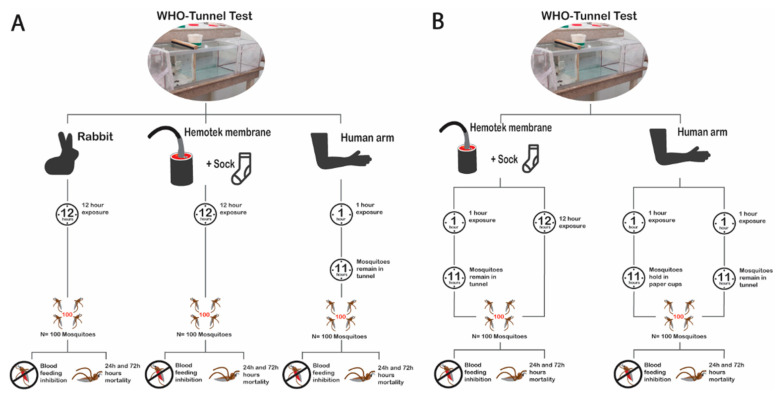
Flow chart of experimental procedure, experiment 1 (**A**—impact of baits) and experiment 2, (**B**—effects of exposure time 12-h vs. 1-h) on WHO tunnel test outcomes.

**Figure 3 insects-13-00562-f003:**
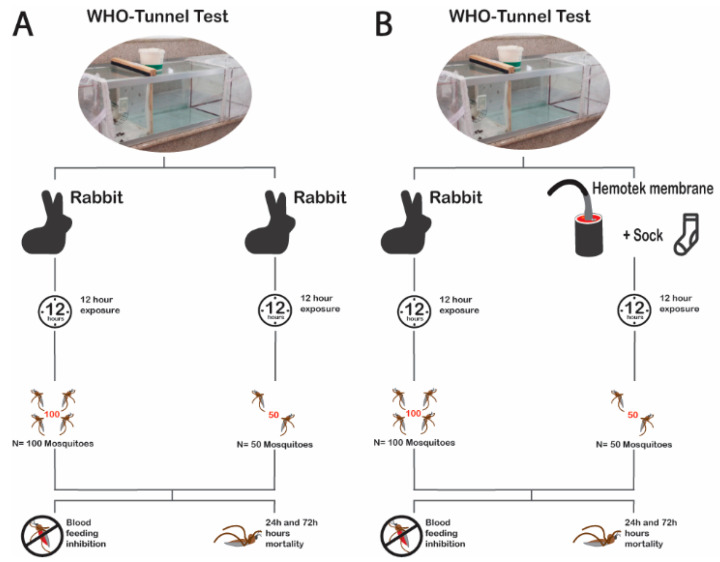
Flow chart of experimental procedure: Experiment 3 (**A**—effects of mosquito density 100 vs. 50) and Experiment 4 (**B**—possibility to replace 100-rabbit bioassay with 50-Hemotek membrane).

**Figure 4 insects-13-00562-f004:**
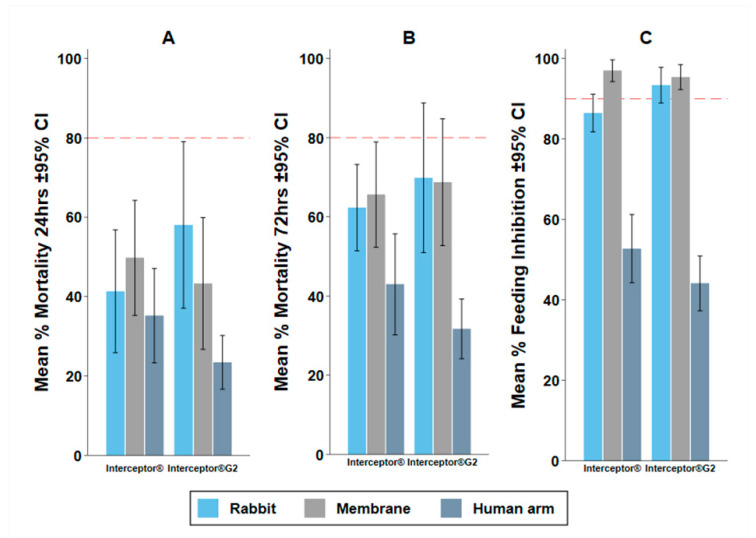
Mean percentage mortality and 95% Confidence Interval (CI) for mortality at (**A**) 24-h (M24), (**B**) 72-h (M72) post exposure and (**C**) blood feeding inhibition (BFI) for Interceptor^®^ and Interceptor^®^ G2 nets with 100 pyrethroid-resistant *Anopheles arabiensis* mosquitoes using rabbit, Hemotek^®^ membrane feeders and human arm as bait in the WHO tunnel bioassay. Red dashed line depicts the WHO minimum bioefficacy criteria of ≥80% M24 and ≥95% BFI.

**Figure 5 insects-13-00562-f005:**
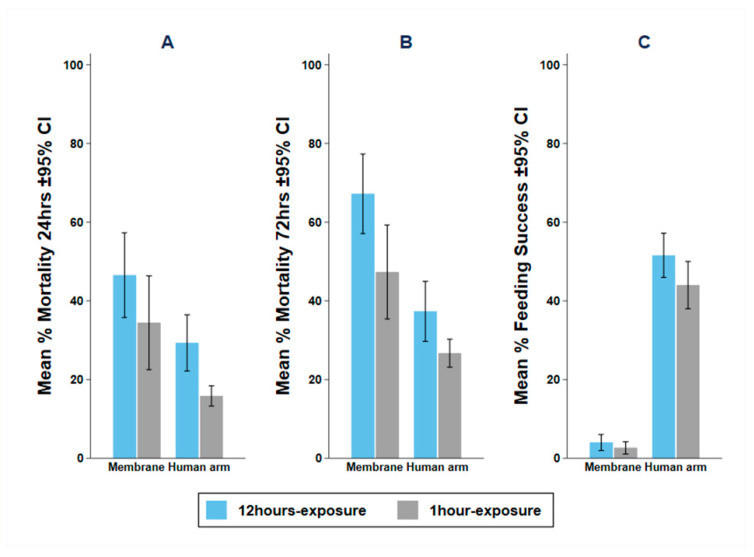
Mean and 95% Confidence Interval (CI) for (**A**) 24-h (M24); (**B**) 72-h (M72); and (**C**) blood feeding success (BFS) with 100 pyrethroid-resistant *Anopheles arabiensis* mosquitoes with 12 h or 1 h exposure time in the WHO tunnel bioassay using Hemotek^®^ membrane or human arm as bait.

**Figure 6 insects-13-00562-f006:**
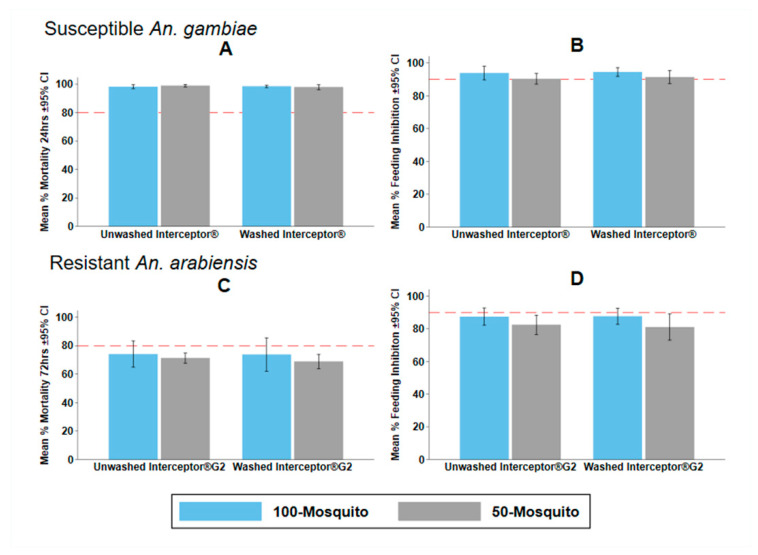
Mean and 95% confidence Interval (CI) for (**A**) 24 h mortality (M24), (**B**) blood feeding inhibition (BFI) of Interceptor^®^ ITN with 100 vs. 50 pyrethroid susceptible *Anopheles gambiae*; (**C**) 72-h mortality (M72); and (**D**) BFI of Interceptor^®^ G2 ITN with 100 vs. 50 pyrethroid-resistant *Anopheles arabiensis* in the WHO tunnel test. Red dashed line depicts WHO minimum bioefficacy thresholds of ≥80% M24 and ≥95% BFI.

**Figure 7 insects-13-00562-f007:**
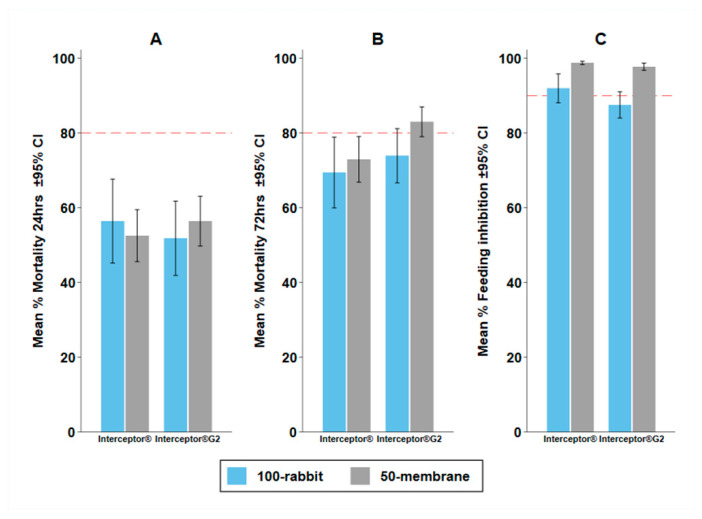
Mean percentage mortality and 95% confidence interval (CI) for (**A**) 24-h (M24); (**B**) 72-h (M72); and (**C**) blood feeding inhibition (BFI) for Interceptor^®^ and Interceptor^®^ G2 nets against pyrethroid-resistant *Anopheles arabiensis* with 100-rabbit (rabbit bait and density of 100 mosquitoes) and 50-membrane (Hemotek^®^ membrane bait and density of 50 mosquitoes) in the WHO tunnel test. The red dashed line depicts the WHO minimum bio-efficacy thresholds of ≥80% M24 and ≥95% BFI.

**Table 1 insects-13-00562-t001:** Experimental design.

Experiment	1	2	3		4
Factor	Host/Baits	Exposure Time	Mosquito Density		Replacement of Rabbit
Comparison	Human or membrane vs. rabbit with 100 mosquitoes	1 h vs. 12 h for human or membrane (within host)	50 vs 100 mosquitoes using rabbit		Rabbit with 100 mosquitoes vs. membrane with 50 mosquitoes
ITNs arms	Interceptor^®^ G2 Unwashed Interceptor^®^ G2 Washed 20× Interceptor^®^ Unwashed Interceptor^®^ Washed 20× Negative control		Interceptor^®^ G2 Unwashed Interceptor^®^ G2 Washed 20× Negative control	Interceptor^®^ Unwashed Interceptor^®^ Washed 20× Negative control	Interceptor^®^ G2 Unwashed Interceptor^®^ G2 Washed 20× Interceptor^®^ Unwashed Interceptor^®^ Washed 20× Negative control
Replicates per arm per comparison	5		15	15	15
Total replicates	75	100	90	90	150
Number of nights	15	10	10	10	16
Mosquitoes exposed	100		100, 50		100, 50
Host/bait	Rabbit, Human, Membrane	Human, Membrane	Rabbit		Rabbit-100, Membrane-50
Exposure time	12 h	12 h 1 h	12 h		12 h
Mosquito species	Anopheles arabiensis		Anopheles arabiensis	Anopheles gambiae	Anopheles arabiensis
Primary Outcomes	Blood feeding success (BFS), 24-h mortality (M24), 72-h mortality (M72)				
Additional Outcome	Blood feeding Inhibition (BFI)				

**Table 2 insects-13-00562-t002:** Impact of bait on mortality and blood-feeding adjusted for the net condition. The difference in the odds of mosquito mortality at 24-h (M24) or 72-h (M72) and blood feeding success (BFS) for 100 pyrethroid-resistant *Anopheles arabiensis* exposed to Interceptor^®^ and Interceptor^®^ G2 with either a rabbit, human arm or membrane feeder as bait *.

		BFS		M24				M72	
	% (95% CI)	OR (95%CI)	*p*-Value	% (95% CI)	OR (95%CI)	*p*-Value	% (95% CI)	OR (95%CI)	*p*-Value
**Control**									
**Rabbit**	64.8 (51.2–78.3)	1		3.8 (0.8-6.8)	1		7.7 (5.1–10.3)	1	
**Membrane**	22.8 (10.4–35.1)	0.16 (0.14–0.20)	<0.001	6.8 (5.9–7.6)	1.83 (1.22–2.75)	0.004	8.9 (8.3–9.5)	1.16 (0.84–1.59)	0.366
**Human arm**	74.4 (67.9–80.8)	1.59 (1.25–2.02)	<0.001	6.4 (4.9–7.8)	1.71 (1.05–2.77)	0.030	11.7 (9.0–14.4)	1.58 (1.11–2.26)	0.012
**Treatment**									
**Rabbit**	6.6 (2.2–11.0)	1		49.7 (36.4–62.9)	1		66.1 (55.3–76.9)	1	
**Membrane**	4.6 (1.5–7.7)	0.34 (0.28–0.48)	<0.001	46.5 (35.7–57.3)	0.90 (0.79–1.02)	0.086	67.2 (57.0–77.3)	1.07 (0.93–1.22)	0.352
**Human arm**	55.9 (49.1–62.7)	9.81 (8.25–11.67)	< 0.001	29.3 (22.1–36.5)	0.42 (0.37–0.48)	<0.001	37.3 (29.7–45.0)	0.31 (0.27–0.35)	<0.001

* Mosquitoes were exposed for 12 h. Data presented are mean proportion (%) with 95% confidence interval (95% CI) and odds ratios (OR) derived from regression analysis with 95% CI adjusted for net type and condition.

**Table 3 insects-13-00562-t003:** Impact of exposure time on mortality and blood-feeding adjusted for the net condition; The difference in the odds of mosquito mortality at 24-h (M24) or 72-h (M72) and blood-feeding success (BFS)) for 100 pyrethroid-resistant *Anopheles arabiensis* exposed to Interceptor^®^ and Interceptor^®^ G2 with either a human arm or a membrane feeder as bait *.

Assays	BFS			M24				M72	
	%	OR	*p*-Value	%	OR	*p*-Value	%	OR	*p*-Value
	(95% CI)	(95% CI)		(95% CI)	(95% CI)		(95% CI)	(95% CI)	
**Membrane**									
1 h-exposure	1.2 (0.1–2.3)	1		24.7 (17.0–32.4)	1		38.9 (26.5–51.2)	1	
12 h-exposure	4.6 (1.5–7.7)	1.55 (1.08–2.22)	0.017	43.3 (25.9–60.6)	1.66 (1.46–1.89)	<0.001	68.8 (52.0–85.5)	2.30 (2.02–2.62)	<0.001
**Human arm**									
1 h-exposure	NA			20.3 (17.7–22.8)	1		31.1 (26.1–36.1)	1	
12 h-exposure	NA			35.2 (22.7–47.6)	2.26 (1.93–2.64)	<0.001	43.0 (29.6–56.3)	1.66 (1.45–1.90)	<0.001

* Mosquitoes were exposed for either 1 h before being removed from the tunnel and placed in holding cups with access to sugar or left overnight in the tunnel for 12 h. Data presented are a mean proportion (%) with a 95% confidence interval (95% CI) and odds ratios (OR) derived from regression analysis with 95% CI adjusted for net conditions.

**Table 4 insects-13-00562-t004:** Effects of mosquito density on mortality and blood-feeding. The difference in the odds of mosquito mortality at 24 h (M24) or 72 h (M72) and blood feeding success (BFS) for resistant *Anopheles arabiensis* exposed to Interceptor^®^ G2 or susceptible *Anopheles gambiae* to Interceptor^®^ in the gold standard rabbit-100 and 50-rabbit mosquitoes *.

	BFS			M24			M72		
Density	%	OR	*p*-Value	%	OR	*p*-Value	%	OR	*p*-Value
	(95% CI)	(95% CI)		(95% CI)	(95% CI)		(95% CI)	(95% CI)	
**Susceptible *An. gambiae* with Interceptor^®^**									
100 Mosquitoes	5.8 (3.4–8.2)	1		98.3 (97.5–99.1)	1		99.1 (98.6–99.6)	1	
50 Mosquitoes	9.1 (6.6–11.6)	2.35 (0.80–6.92)	0.122	98.4 (97.5–99.3)	1.10 (0.32–3.72)	0.882	99.6 (99.3–99.9)	1.80 (0.43–7.54)	0.421
**Resistant *An. arabiensis* with Interceptor^®^ G2**									
100 Mosquitoes	12.5 (8.9–16.0)	1		51.8 (41.9–61.7)	1		73.9 (66.7–81.2)	1	
50 Mosquitoes	18.3 (13.3–23.2)	1.54 (0.74–3.22)	0.249	45.1 (40.7–49.6)	0.69 (0.23–2.12)	0.518	70.0 (67.0–73.1)	0.65 (0.25–1.67)	0.375

* Mosquitoes were exposed for 12 h in the tunnel. Data presented are a mean proportion (%) with 95% confidence interval (95% CI) and odds ratios (OR) derived from regression analysis with 95% CI adjusted for net type and condition.

**Table 5 insects-13-00562-t005:** **Superiority of Interceptor^®^ G2 over Interceptor^®^ using 100 versus 50 resistant mosquitoes:** The difference in the odds of mosquito at 24 h (M24) and 72 h (M72) and blood feeding success (BFS) for pyrethroid-resistant *Anopheles arabiensis* exposed to Interceptor^®^ G2 and Interceptor^®^ in the gold standard rabbit-100 and 50-rabbit mosquitoes *.

Treatment	100-Rabbit				50-Rabbit			
	BFS		M72		BFS		M72	
	OR (95% CI)	*p*-Value	OR (95% CI)	*p*-Value	OR (95% CI)	*p*-Value	OR (95% CI)	*p*-Value
**Overall**								
Interceptor^®^	1		1		1		1	1
Interceptor^®^ G2	1.76 (1.55–1.99)	<0.001	1.23 (1.13–1.33)	<0.001	12.93 (9.63–17.36)	<0.001	1.41 (1.26–1.57)	<0.001
**Unwashed**								
Interceptor^®^	1		1		1		1	1
Interceptor^®^ G2	1.64 (1.38–1.95)	<0.001	1.15 (1.02–1.29)	0.018	8.50 (5.95–12.15)	<0.001	1.83 (1.56–2.14)	<0.001
**Washed 20×**								
Interceptor^®^	1		1		1		1	1
Interceptor^®^ G2	1.90 (1.58–2.27)	<0.001	1.31 (1.17–1.47)	<0.001	24.34 (14.16–41.85)	<0.001	1.07 (0.85–1.34)	0.432

* Mosquitoes were exposed for 12 h in the tunnel. Data presented are a mean proportion (%) with a 95% confidence interval (95% CI) as well as odds ratios (OR) derived from regression analysis with 95% CI, adjusted for net conditions.

**Table 6 insects-13-00562-t006:** Comparison of the membrane assay to the gold standard with rabbit assay. The difference in the odds of mosquito mortality at 24-h (M24) and 72-h (M72) and blood feeding success (BFS) for resistant *Anopheles arabiensis* was measured between the gold standard rabbit assay with 100 mosquitoes and the membrane assay with 50 mosquitoes *.

Assay	BFS		*p*-Value	M24		*p*-Value	M72		*p*-Value
	% (95% CI)	OR (95% CI)		% (95% CI)	OR (95% CI)		% (95% CI)	OR (95% CI)	
**Interceptor^®^**									
100 Rabbit	7.9 (4.1–11.8)	1		56.4 (45.3–67.6)	1		69.4 (60.0–78.8)	1	
50 Membrane	1.2 (0.8–1.7)	0.19 (0.08–0.45)	<0.001	52.5 (45.6–59.4)	0.39 (0.10–1.61)	0.195	73.0 (66.9–79.0)	0.54 (0.14–2.06)	0.370
**Interceptor^®^ G2**									
100 Rabbit	12.5 (9.0–16.0)	1		51.8 (42.0–61.7)	1		73.9 (66.7–81.1)	1	
50 Membrane	2.3 (1.3–3.2)	0.17 (0.09–0.30)	<0.001	56.4 (49.8–63.1)	1.10 (0.51–2.36)	0.814	83.0 (79.1–86.9)	1.50 (0.75–2.98)	0.251

* Mosquitoes were exposed for 12 h in the tunnel. Data presented are a mean proportion (%) with a 95% confidence interval (95% CI) as well as odds ratios (OR) derived from regression analysis with a 95% CI adjusted for net type.

**Table 7 insects-13-00562-t007:** Superiority of Interceptor^®^ G2 over Interceptor^®^ was estimated by comparing the membrane assay to the gold standard assay with pyrethroid-resistant mosquitoes. The difference in the odds of mosquito at 72-h (M72) and blood feeding success (BFS) for resistant *Anopheles arabiensis* measuring the superiority of Interceptor^®^ G2 and Interceptor^®^ with the gold standard with 100-rabbit compared to 50-membrane bioassays *.

Treatment	100-Rabbit				50-Membrane			
	BFS		M72		BFS		M72	
	OR (95% CI)	*p*-Value	OR (95% CI)	*p*-Value	OR (95% CI)	*p*-Value	OR (95% CI)	*p*-Value
**Overall**								
Interceptor^®^	1		1		1		1	
Interceptor^®^ G2	1.76 (1.47–2.10)	<0.001	1.23 (1.10–1.38)	<0.001	1.87 (1.05–3.33)	0.033	1.79 (1.50–2.14)	<0.001
**Unwashed**								
Interceptor^®^	1		1		1		1	
Interceptor^®^ G2	1.64 (1.28–2.09)	<0.001	1.15 (0.98–1.35)	0.094	2.34 (1.11–4.93)	0.025	1.81 (1.43–2.29)	<0.001
**Washed 20×**								
Interceptor^®^	1		1		1		1	
Interceptor^®^ G2	1.90 (1.47–2.45)	<0.001	1.31 (1.12–1.54)	0.001	1.26 (0.49–3.20)	0.634	1.82 (1.39–2.37)	<0.001

* For the gold standard, 100-mosquitoes with rabbit and 50-mosquito with 2 Hemotek^®^ membrane feeders augmented with worn socks were used in the WHO tunnel bioassay, adjusted for net type and condition.

## Data Availability

The data set for this study is available on reasonable request from Vector Control Product and Testing Unit of Ifakara Health Institute.

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
