# Peer review of "Modified World Health Organization (WHO) Tunnel Test for Higher Throughput Evaluation of Insecticide-Treated Nets (ITNs) Considering the Effect of Alternative Hosts, Exposure Time, and Mosquito Density"

_insects, 2022, doi:10.3390/insects13070562_

Round 1

Reviewer 1 Report

  1. Fig should be black and white so every (color blind) one can easily get that 
  2. Cross-check the results with tables, please 
  3. Add some data in the introduction portion, please.
  4. If possible then please add the ethical letter in supplementary material as only ref No. is not enough in such kind of study.
  5. Mention the Generation No of Mosquitoes used in experimentation

Author Response

See the file.

Author Response

See the file.

Reviewer 3 Report

The manuscript describes a very comprehensive set of experiments evaluating modifications to the standard WHO tunnel test.  Performing 5 tunnels a night so that controls and treatments could be run simultaneously was a strength of the study. The authors convincingly show that halving the number of mosquitoes in each test would enable a higher number of nets to be tested without compromising the value of the data. They also show some promising initial results with replacing the animal bait with an artificial membrane but further optimisation is needed before this method could be considered a suitable alternative to an animal.  In contrast, results with a human arm showed great promise but were surprisingly discounted by the authors as they were too dissimilar to the results with the rabbit. 

Major Edits

1.       The sample size description (lines 268-272) appears to be from another manuscript.

2.       Table 1 is missing

3.       The importance of consistent blood feeding rates in the controls is stressed in the text of the manuscript (line 385, line 508) but the data from these experiments is buried in supplementary data and all data in main body of the paper is presented as OR. Given the very low rates of blood feeding in the membrane experiments this data needs to be in the main tables themselves.  It would also be interesting to see if there was a consistent effect of host source on control blood feeding rates (e.g was blood feeding success consistently higher in human arm vs rabbit?  And rabbit vs membrane).

Recommended Edits

4.       One of the issues with use of the human arm was the very high level of blood feeding, which resulted in a lower mortality.  One possible modification might be to use a human arm but with a barrier that prevented feeding (assuming the outcome of interest is mortality, rather than blood feeding inhibition). Further discussion about the confounding effect of the two outcomes from tunnel tests, and which outcome is of most relevant would be an interesting addition to the discussion.  Similarly, much is made about the congruence between experimental hut studies and tunnel tests with rabbits. But, as far as I am aware, there have been on studies to see if the correlation of tunnel tests using human arms EHTs would be stronger.

Minor points

5.       State the blood source used for mosquito rearing – was it cow, as used in the experiments?

6.       Provide further information on the Kingani strain or provide a reference to support overexpression of P450s and restoration of susceptibility by PBO

7.       Line 167 spelling ‘post’ not ‘most’

8.       Consider rephrasing sentence on line 228-232 as it is difficult to follow

9.       Figure 3B – should this show 100-rabbit (not 50)?

10.   The p value of <0.001 comparing the BFS between rabbit and membrane in Table 2 is surprising given the confidence intervals – the values also differ slightly in the text below (OR 0.37 vs 0.34) so these need checking

Author Response

See the file.

Reviewer 4 Report

Duplicate data presented in most figures and tables in the Results section. All comments and suggestions are provided in the attached file

Author Response

See the file.
